# New Developments in Carbonic Anhydrase IX-Targeted Fluorescence and Nuclear Imaging Agents

**DOI:** 10.3390/ijms23116125

**Published:** 2022-05-30

**Authors:** Kuo-Ting Chen, Yann Seimbille

**Affiliations:** 1Department of Chemistry, National Dong Hwa University, Hualien 974301, Taiwan; 2Department of Radiology and Nuclear Medicine, Erasmus MC, University Medical Center Rotterdam, Wytemaweg 80, 3015 CN Rotterdam, The Netherlands; y.seimbille@erasmusmc.nl; 3Erasmus MC Cancer Institute, Erasmus University Medical Center, Doctor Molewaterplein 40, 3015 GD Rotterdam, The Netherlands

**Keywords:** carbonic anhydrase IX, cancer, imaging agents, PET, SPECT, fluorescence imaging

## Abstract

Carbonic anhydrase IX (CAIX) is a tumor-specific and hypoxia-induced biomarker for the molecular imaging of solid malignancies. The nuclear- and optical-imaging of CAIX-expressing tumors have received great attention due to their potential for clinical applications. Nuclear imaging is a powerful tool for the non-invasive diagnosis of primary and metastatic CAIX-positive tumors and for the assessment of responses to antineoplastic treatment. Intraoperative optical fluorescence imaging provides improved visualization for surgeons to increase the discrimination of tumor lesions, allowing for safer surgical treatment. Over the past decades, many CAIX-targeted molecular imaging probes, based on monoclonal antibodies, antibody fragments, peptides, and small molecules, have been reported. In this review, we outline the recent development of CAIX-targeted probes for single-photon emission computerized tomography (SPECT), positron emission tomography (PET), and near-infrared fluorescence imaging (NIRF), and we discuss issues yet to be addressed.

## 1. Introduction

### 1.1. CAIX Is a Biomarker for the Detection of Solid Tumors

In the clinical setting, tumor hypoxia is usually considered to be a negative therapeutic indicator. Hypoxia is a central environmental feature of solid tumors having a low level of oxygen, usually caused by poor perfusion from an irregular and inefficient vasculature. Tumor cells respond to hypoxic microenvironmental stress by reprogramming their metabolisms to engage less efficient glycolytic pathways without the need of oxygen, also known as the Warburg effect. Adapting to this type of glycolysis results in the production of acidic metabolites and leads to a decline in extracellular pH, creating a toxic microenvironment for the cancer cells. CAIX is a zinc metalloprotein enzyme belonging to the carbonic anhydrase family. Under acidic conditions, CAIX is mediated by the hypoxia-inducible factor 1-α (HIF1-α) and is strongly upregulated in hypoxic tumors. The overexpression of CAIX contributes to cancer cell survival by catalyzing the interconversion of carbon dioxide (CO_2_) and bicarbonate (HCO_3_^−^) to maintain intracellular pH homeostasis [1,2]. In addition to the modulation of the microenvironment, CAIX also plays an important role in mediating tumor progression, invasion, and metastasis [3,4]. Thus, CAIX is an attractive target for cancer diagnosis and therapy for the following reasons: (1) CAIX expression in tumors is upregulated and associated with poor prognosis; (2) CAIX is overexpressed in a wide variety of solid tumors, such as renal cell carcinoma, brain, bladder, cervical, head and neck, breast, lung, and kidney cancers; (3) CAIX expression in normal tissues is highly restricted; (4) CAIX is easily accessible for drug targeting due to its location on the cell membrane. As a result, non-invasive imaging techniques could be used to detect CAIX, to identify hypoxic areas in solid tumors, and to determine the most appropriate treatment regimens for cancer patients. Notably, overexpression of CAIX is found in clear cell renal cell carcinoma (ccRCC) under normoxia conditions. Due to the Von Hippel–Lindau (VHL) mutation of ccRCCs, HIF1-α is overexpressed, leading to the overexpression of CAIX on the ccRCC cell surface [5]. Imaging CAIX is a promising approach to detecting ccRCC. 

### 1.2. Fluorescence Imaging and Nuclear Imaging of CAIX

Fluorescence imaging is a widely used tool allowing for the visualization of drug location, gene expression, and enzyme activity in living systems. In the early days, CAIX-targeted molecules were labeled with a visible light fluorescence dye (Em = 400~700 nm) to study the correlation between cellular CAIX overexpression and hypoxia level [6]. However, such dye is limited to in vitro studies due to its low tissue-penetration and autofluorescence. NIRF imaging addresses these limitations and has emerged as a powerful tool for in vivo studies. NIRF imaging can detect fluorescent agents approximately 2 to 4 cm deep in the tissue with emission wavelengths ranging from 650 to 900 nm [7]. It greatly facilitates in vivo applications, such as fluorescence-guided surgery. The applications of NIRF-labeled probes include aiding surgeons in delineating surgical tumor margins and performing resections more safely. The clinically approved indocyanine green (ICG) has been widely used in various CAIX ligands to detect CAIX-positive tumors. The current trend in probe development is to label CAIX-ligands with the brighter and longer wavelength NIRF dye to increase imaging contrast. However, the detection of deeply buried tumors and distant metastases is still limited by the short-penetration fluorescence. Nuclear imaging, such as PET and SPECT, associated with CAIX-targeted radiotracers could provide non-invasive, quantitative, and highly sensitive whole-body imaging of tumor hypoxia. Currently, many radiolabeled, CAIX-targeted antibodies have advanced to clinical trials, but only a few small-molecule-based radioligands have been tested in a clinical setting, and most of them remain in the preclinical stage. The major challenge of low-molecular-weight probe development is to improve their stability, target specificity, and pharmacokinetic properties, especially their ability to discern CAIX from other CAs. In this review, we focused on NIRF- and nuclear imaging CAIX-targeted imaging agents and highlighted the strategies for enhancing imaging contrast with labeled antibodies, peptides and small-molecule-based probes.

## 2. Antibody-Based Imaging Agents

Monoclonal antibodies (mAb) are widely used in the field of imaging because of their excellent target specificity and low uptake in nontarget tissues. The high molecular weight of mAbs allows labeling with dyes and radiometal–chelator complexes without significant loss of their immunoreactivity. However, it may lead to suboptimal penetration into solid tumors. G250 and its chimeric form cG250 (girentuximab) are the most clinically investigated CAIX-targeted mAbs. They exhibit high affinity to the extracellular proteoglycan (PG)-like domain of the enzyme, and the PG-like domain only exists in CAIX. They both have high selectivity toward CAIX over other CA isomers [8]. Many radiolabeled girentuximab derivatives have been reported for imaging and treatment of VHL-mutation-induced ccRCC. In 1993, Oosterwijk et al. first reported a phase I study with [^131^I]I-labeled G250 [9]. In the 16 preoperative patients who received 8.4~12.9 mCi of [^131^I]I-labeled G250 at 5 dose levels, 12 CAIX-positive patients positively responded to the treatment, but one CAIX-negative patient showed a false-positive result. The radiotracer uptake by CAIX-positive tumors was observed in a broad range (0.02 to 0.12% ID/g). In PET imaging studies, Divgi et al. labeled the chimeric cG250 with the positron-emitter ^124^I and conducted 2 clinical trials to assess its suitability for ccRCC imaging [10]. The chimeric cG250 showed bioactivity similar to that of G250. In a phase I study, 26 patients received a single dose of 185 MBq/10 mg of [^124^I]I-girentuximab 1 week before surgical resection by laparotomy. Patients were followed by PET/CT scanning of the abdomen 3 h prior to the surgery. Their results showed that 15 of 16 ccRCCs were accurately identified by the [^124^I]I-mAb immuno-PET (sensitivity: 94%), and 9 non-ccRCCs (e.g., papillary RCC, angiomyolipoma, etc.) were not detected by the tracer; the negative predictive value (NPV) and positive predictive value (PPV) were 90% and 100%, respectively. A phase III study with 226 preoperative patients reported a sensitivity, PPV, and NPV of [^124^I]I-girentuximab for detecting ccRCC of 86%, 94%, and 70%, respectively. However, the sensitivity was lower for tumor sizes smaller than 2 cm (70.8%) [11]. Parallel to the development of [^124^I]I-girentuximab, another radiometal-labeled [^111^In]In-DTPA-cG250 was evaluated in a clinical setting by Muselaers et al. [12]. A total of 29 patients were enrolled in the trial and received 100–200 MBq of [^111^In]In-DTPA-cG250 4–7 days before SPECT scans were performed. The detection rate of the [^111^In]In-DTPA-cG250 immuno-SPECT was high (PPV = 94%). A total of 15 of 16 cases of ccRCC were successfully identified, and none of the 9 patients with non-ccRCC lesions showed tracer uptake. It is worth noting that, in addition to the detection of primary ccRCC, the [^111^In]In-DTPA-cG250 was able to detect more metastatic lesions than ^131^I-cG250 [13]. Very recently, Merkx et al. reported a phase I study to evaluate the safety, biodistribution, and dosimetry of [^89^Zr]Zr-girentuximab [14]. A total of 10 patients with suspected ccRCC received 37 MBq of [^89^Zr]Zr-girentuximab at mass doses of 5 or 10 mg prior to PET/CT scans at 0.5–168 h post-administration. In most patients, the tumor lesions were visible after 24 h post-administration (PA), and in one case, the tumor could be detected as early as 0.5 h PA. (Figure 1). Overall, the [^89^Zr]Zr-immuno PET was safe and allowed for the successful differentiation between ccRCC and non-ccRCC lesions (6 ccRCC-positive patients; 4 negative scans of non-ccRCC patients). A phase III study with [^89^Zr]Zr-girentuximab is planned to assess the diagnostic accuracy in patients (NCT03849118).

In addition to the radiolabeled girentuximab, Muselaers et al. developed a dual-modality [^125^I]I-girentuximab-IRDye800CW and performed preclinical evaluation in mice bearing CAIX-positive SKRC-52 tumors [15]. The mice were imaged at 1 and 3 d post-injection (PI) of [^125^I]I-girentuximab-IRDye800CW. The results showed an excellent concordance between the micro-SPECT and optical imaging. The tumors were clearly visualized at 24 h PI, and the tumor uptake was 6.9% ID/g at 72 h PI. However, the uptake was lower compared to [^125^I]I-girentuximab (14.8% ID/g, 72 h PI), which may be due to the faster clearance of the IRDye800CW conjugated antibody. Shortly thereafter, the same group performed dual-modality imaging with [^111^In]In-DTPA-G250-IRDye800CW in a ccRCC mouse model [16]. The SPECT and fluorescence images showed good concordance and clear delineation of the ccRCC lesions at 48 h PI. The maximum tumor uptake was high, up to 58.5% ID/g at 168 h PI. The promising results of the preclinical evaluation of this dual-modality probe have led to the initiation of a phase I clinical study to assess its feasibility and safety in ccRCC patients. In 2018, a phase I dose-escalation study of [^111^In]In-DOTA-girentuximab-IRDye800CW in 15 patients with primary renal lesions was performed by Hekman et al. [17] The dual-modality probe (5, 10, 30, or 50 mg) was administered to patients, and SPECT/CT was performed at 4 d PI. The probe showed high sensitivity; all CAIX-expressing tumors were visualized by SPECT/CT, while no uptake was observed in CAIX-negative tumors. Excellent concordance between the SPECT/CT and the NIRF imaging was also observed. During surgery, all ccRCC could be localized by gamma probe measurements with a mean T:N ratio of 2.5 at all protein doses. The optimal T:N ratio (3.3) was found at the 10 mg protein dose, but in general, the protein dose did not significantly affect the T:N ratio (average: 2.5). The phase I study showed that [^111^In]In-DOTA-girentuximab-IRDye800CW is safe for CAIX-targeted, dual-modality imaging, and it can be used for intraoperative guidance of ccRCC resection.

However, full-size monoclonal antibodies have relatively poor tumor penetration and slow blood clearance, thus requiring a long time to reach the best signal-to-background ratio. In radio imaging, radiation dose accumulation due to the prolonged circulation of the radiotracers in the body could cause a burden for the patients. Antibody fragments with lower molecular weights tend to have rapid localization in tumors. Various types of antibody fragments, such as F(ab’)_2_ (~100 kD), nanobody (~15 kD), and affibody (~7 kD), have been developed to overcome this circumstance (Table 1). Hoeben et al. reported the use of [^89^Zr]Zr-DFO-cG250-F(ab’)_2_ as a PET marker of hypoxia in a head and neck xenograft tumor model (SCCNij-3) [18]. The cG250-F(ab’)_2_ was prepared using an enzymatic digestion method. The maximal tumor uptake (3.71% ID/g) of the tracer was found at 4 h PI, but normal tissues (blood, kidney, and liver) also exhibited high uptake of the tracer. The [^89^Zr]Zr-cG250-F(ab’)_2_ demonstrated specific imaging of CAIX-expressing tumors. Prompted by the results, Huizing et al. performed a preclinical validation of the [^111^In]In-labeled cG250-F(ab’)_2_ [19]. The specific tracer accumulation (4.1% ID/g) was reached at 24 h PI, and the tumor-to-blood (T/B) and tumor-to-muscle (T/M) ratios of [^111^In]In-DTPA-cG250-F(ab’)_2_ were 30.8 and 12.1, respectively. However, tracer uptake in the kidneys (65% ID/g) was still high, which may be caused by renal tubular reabsorption. Although high kidney uptake is not a limitation in the case of head and neck tumors, it could interfere with the imaging of tumor masses near the kidney. The probe is therefore not optimal for the detection of CAIX-positive renal cancers. Following the previous research, the protein dose, timing, and image acquisition of [^111^In]In-DTPA-cG250-F(ab’)_2_ were optimized for a quantitative SPECT imaging analysis of HNSCCs [20]. A protein dose of 10 μg showed the highest tumor uptake of 3.0% ID/g at 24 PI, which is in line with the quantitative microSPECT imaging results (Pearson correlation coefficient (r) = 0.78). In addition, [^111^In]In-DTPA-cG250-F(ab’)_2_ has been used to monitor the quantitative change in CAIX expression in tumor therapy [21]. Huizing et al. detected a 69% decrease of CAIX levels in a FaDu tumor model after treatment by Atovaquone. However, SPECT imaging with [^111^In]In-DTPA-cG250-F(ab’)_2_ did not discriminate between the treated and control tumors. According to the authors, the uptake by the treated tumors may also contribute to the Atovaquone-induced enhanced permeability and retention effect (EPR), which causes it to fail in the application of therapy monitoring. Using smaller biovectors, such as nanobodies or affibodies, may alleviate the EPR effect. 

Besides G250 and its fragments, newly developed nanobodies were applied to CAIX-targeted probe preparation. In 2016, van Brussel et al. developed the CAIX-specific nanobody VHH-B9 by phage display selection [25]. VHH-B9 was conjugated with an IRDye800W dye (B9-IR) and evaluated in a xenograft breast cancer mouse model using ductal carcinoma in situ cells (DCIS). Tumor uptake of the fluorescent tracer was 14.0% ID/g, and the T/M and T/B ratios were 70 and 23, respectively. An optical imaging study of BR-I9 showed clear visualization of CAIX-positive DCIS tumors within 2 h after probe administration. Moreover, the rapid pharmacokinetics and probe stability might provide better imaging contrast than conventional CAIX-IHC for pathologic assessment. Very recently, van Lith et al. reported the [^111^In]In-labeled VHH-B9 in the absence or presence of an albumin-binding domain (ABD) [22]. The ABD on VHH increased the plasma half-life of the VHH, therefore improving the tumor uptake of the tracer. In the comparison study reported by van Lith, the uptake of [^111^In]In-DTPA-VHH-B9 and [^111^In]In-DTPA-VHH-B9-ABD were 0.51 and 8.7% ID/g, respectively. Not surprisingly, the tumor was only visualized with [^111^In]In-DTPA-VHH-B9-ABD in SPECT/CT images. However, the uptake of [^111^In]In-DTPA-VHH-B9-ABD did not decrease after administration of an excess of VHH, which means that the uptake was not CAIX-specific. The authors concluded that the addition of ABD to B9 did not improve SPECT imaging contrast in head and neck cancer. 

Affibodies are small proteins based on non-immunoglobulin scaffolds, and they have been used in CAIX imaging [26]. In 2019, Huizing et al. performed an in vivo comparison of the affibody-based [^111^In]In-DTPA-ZCAIX:2 and two cG250-based radiotracers in a HNSCC xenograft model [23]. Tracer uptake of [^111^In]In-DTPA-cG250, [^111^In]In-DTPA-cG250-F(ab’)_2_, and [^111^In]In-DTPA-ZCAIX:2 in tumors were 30% ID/g at 72 h PI, 3.0% ID/g at 24 h PI, and 0.32% ID/g at 4 h PI, respectively. The tumors were clearly visualized with [^111^In]In-DTPA-cG250 and [^111^In]In-DTPA-cG250-F(ab’)_2_ at 24 and 72 h PI, respectively, but not visible with [^111^In]In-DTPA-ZCAIX:2. Meanwhile, Garousi et al. reported another comparison study between [^111^In]In-DTPA-cG250-F(ab’)_2_ and [^111^In]In-DTPA-ZCAIX:2 in the ccRCC model (SKRC-52) [24]. Unlike the abovementioned results, the tumor uptake of the affibody-based probe (15% ID/g) was higher than that of the F(ab’)_2_-based probe (6% ID/g) at 4 h PI, and both radiotracers were capable of visualizing tumors at 4 h PI In a SPECT imaging study, the contrast was higher with [^111^In]In-DTPA-ZCAIX:2 than it was with the F(ab’)_2_-based probe. However, the high kidney uptake (392% ID/g) hampers the application of this tracer for the imaging of primary ccRCC tumors, but that does not prevent its use in detecting metastases.

## 3. Peptide-Based Compounds

Peptides are recognized for being highly selective, efficient, and relatively safe vectors. Peptide-based imaging probes typically have a high binding affinity for the target, specific uptake and retention in the target tissue, and rapid clearance from non-target organs. A significant number of peptides, such as cyclic RGD peptides, somatostatin (SST), gastrin-releasing peptide (GRP), glucagon-like peptide-1 (GLP-1), and neuropeptide-Y (NPY), have been labeled with a wide range of imaging moieties for use as in vivo imaging probes. However, research on CAIX-targeted peptides is still limited. In 2010, Askoxylakis et al. identified a dodecapeptide CaIX-P1 (YNTNHVPLSPKY) that targets the extracellular domain of CAIX via a phage display method [27]. The CaIX-P1 contains a *N*-terminal tyrosine residue that was radiolabeled with ^125/131^I, and the radiopeptide was evaluated in vitro and in vivo for binding affinity, specificity, and biodistribution. The binding of [^125^I]I-CaIX-P1 on CAIX-positive SKRC52, HCT116, and HT-29 cells showed a correlation between cell uptake and CAIX expression, but the IC_50_ values, in the micromolar range, were not satisfying. Biodistribution studies of ^131^I-CaIX-P1 were performed in SKRC-52 and HCT-116 xenografted mice [27,28]. In both experiments, the T/B ratio was found to be lower than 1 at all time points. The highest contrast in SKRC52 and HCT116 were achieved at 1 h and 2 h PI, respectively, and the tumor uptakes of the tracer decreased over time. The low tumor uptakes could be attributed to the low binding of CaIX-P1 to CAIX and the in vivo peptide degradation (half-life of CaIX-P1: 25 min). Besides, no imaging study of the ^125/131^I-CaIX-P1 was presented in the report. 

To improve the stability and binding properties of the CAIX-targeted peptide, Rana et al. took advantage of the alanine-scanning method to optimize the amino acid sequence of CaIX-P1. It resulted in the new peptide, CaIX-P1-4-10 (NHVPLSPy) [29]. [^125^I]I-CaIX-P1-4-10 exhibited a 5.8-fold higher binding affinity than [^125^I]I-CaIX-P1, and it was stable in serum for 90 min. However, tumor uptake of ^131^I-CaIX-P1-4-10 was not significantly improved (~2.5% ID/g at 1 h PI), and SKRC-52 tumors could not be distinguished from the background. To improve the isoform selectivity, Rena et al. identified a new linear dodecapeptide PGLR-P1 (NMPKDVTTRMSS), which targets the region of the extracellular proteoglycan (PG)-like domain of CAIX with no homology to other Cas [30]. The [^125^I]I-labeled PGLR-P1 showed a higher selectivity toward CAIX (1.8% applied dose) and the PGLR domain of CAIX (10% applied dose) than CAII and CAXII (both < 0.1% applied dose). However, a rapid peptide degradation was observed with a serum half-life of approximately 20 min. Although the authors attempted to improve the stability by transforming the l-peptide to its d-enantiomer, a complete loss of binding affinity was noticed with the modified peptides. In addition, the uptake of [^125^I]I-PGLR-P1 in SKRC-52 tumor (0.48 ± 0.20% ID/g at 1 h PI) was lower than in most normal tissues. Recently, Jia et al. radiolabeled CaIX-P1-4-10 with ^18^F via a Cu(I)-catalyzed alkyne–azide cycloaddition (CuAAC) [31]. [^18^F]F-CaIX-P1-4-10 (**1**) was obtained in an overall radiochemical yield of 35–45% from aqueous [^18^F]fluoride and >99% radiochemical purity in 70–80 min. MicroPET/CT scans of HT-29-bearing mice allowed for the visualization of the tumor at 1 h PI (Figure 2). The tracer accumulated in the tumor with a mean standardized uptake value (SUV_mean_) of 0.38 ± 0.03, but the uptake in non-target organs remained high (SUV_means_ of liver and kidneys were 14.21 ± 3.68 and 2.08 ± 0.09, respectively). The high background might be due to the rapid degradation of the tracer in serum (48.5% intact after 3 h).

## 4. Small-Molecule-Based Compounds

The most diverse and largest class of CAIX-targeted imaging probes is small-molecule-based compounds. Small-molecular CA inhibitors exhibit high target affinity and a short blood half-life, making them attractive candidates for use as imaging agents. Several classes of small molecules are known as effective CA inhibitors, such as sulfonamides, coumarins/sulfocoumarins, phenols, and dithiocarbamates [32,33,34,35]. Among them, the aromatic sulfonamides, such as benzenesulfonamide (BSA), acetazolamide (AAZ), and imidazothiadiazole sulfonamide (IS), have particularly gained great interest in the development of imaging agents. The inhibition mechanism of the sulfonamides is by coordination of zinc ions within the active site; however, the similarity of the active sites of all CA isomers hampers the selectivity of the probes. Therefore, study on the effects of the diverse structures of CA inhibitors on isoform selectivity is emphasized. Herein, we summarize the current progress of CAIX-targeted molecular imaging probes focused on aromatic sulfonamides (Table 2).

### 4.1. Benzenesulfonamide (BSA)

BSA is one of the earliest-reported CAIX inhibitors used for imaging probe development (Figure 3). In 2005, Cecchi et al. demonstrated that fluorescein-labeled BSAs selectively accumulated in hypoxic cells expressing CAIX with high affinity (*K*_i_ = 24~35 nM) [64]. Encouraged by the positive in vitro results, the in vivo evaluation of the fluorescent-BSA **2** in HT-29 tumors was performed by Dubois et al. A three-fold higher probe accumulation in the tumor site under hypoxia conditions was found compared to the normoxia model [36]. The tumor was visualized in the CAIX-positive model 2 h after injection of the fluorescent sulfonamide. For SPECT imaging, various chelator-containing BSAs have been described. Akurathi et al. reported the synthesis of [^99m^Tc]Tc-labeled BSA **3** using a *N*-2-picolyl-*N*-acetic acid for the complexation with tricarbonyl [^99m^Tc]Tc(I) [37]. High and sustained uptake of the [^99m^Tc]Tc-labeled BSA was observed in the kidney, intestine, and liver (24.2, 29.1, and 38.9% ID/g at 0.5 PI, respectively), but low uptake in the HT-29 tumor (0.2% ID/g at 0.5 PI). The low tumor-to-background ratio suggested that this compound is not a promising tracer for the visualization of CAIX-expressing tumors. The in vitro binding affinity of the BSA-based probe was manipulated by changing the chelator. Lu et al. reported [^99m^Tc]Tc-3d (**4**) using an imidazole-based [^99m^Tc]Tc chelator, showing high affinity (IC_50_ = 9 nM) to CAIX, but no further animal imaging study was presented [38]. Nakai et al. used dipyridylamine and iminodiacetate as chelators to append to sulfonamide (**5** and **6**) for imaging CAIX expression [39]. Although compound **6** showed better binding affinity and higher selectivity than **5**, the tumor uptake of these [^99m^Tc]Tc-tracers was very low (<0.2% ID/g). On the contrary, accumulation of the tracers was high in normal tissues.

In addition to the SPECT tracers, PET tracers based on [^18^F]F- and [^68^Ga]Ga-labeled BSAs have recently gained the most attention. [^18^F]F-VM4-037 was the first small-molecule, CAIX-targeting PET tracer entering phase II clinical trial for ccRCC imaging. Nevertheless, its pronounced uptake in healthy kidneys is a major challenge for the detection of primary ccRCC lesions with this radiotracer. Despite that, [^18^F]F-VM4-037 may be useful in the evaluation of metastatic ccRCC cancer [65]. An early study on a BSA-based PET tracer was reported by Lin and coworkers in 2014. They synthesized and evaluated [^18^F]F-U-104 (**7**) for the detection of CAIX-expressing in HT-29 tumor-bearing mice [40]. The [^18^F]F-FEC (coumarin-based) was also included in this study. The biodistribution of both compounds showed moderate to low tumor uptake (1.16 and 0.83% ID/g for [^18^F]F-FEC and [^18^F]F-U-104, respectively). The authors only performed an imaging study with [^18^F]F-FEC in HT-29 tumor-bearing mice because the T/B ratio of [^18^F]F-U-104 was not favorable. Nevertheless, the tumors could not be visualized on the PET images after injection of [^18^F]F-FEC. 

Considering that isoform-selective probes of CAIX may enable better PET imaging outcomes, Lau and co-workers reported zwitterionic [^18^F]F-labeled BSAs [41]. Low membrane permeability and reduced accumulation in CAII-rich blood cells was expected. The zwitterionic probe [^18^F]F-AmBF_3_-ABS (**8**) presented good CAIX/CAII selectivity, which was nine-fold higher in the stopped-flow CO_2_ hydration assay. Biodistribution and PET imaging were performed at 1 h PI in mice bearing HT-29 tumor xenografts. The probe localized in the tumor (0.64% ID/g at 1 h PI), and tumors could be visualized on the PET images despite the tumor-to-background ratios being substantially low. Based on the same concept, Zhang et al. described the [^18^F]F-labeled cationic sulfonamide derivative **9**, which bears a quaternary ammonium group [42]. Compound **9** exhibited a *K*_i_ value of 0.22 μM for CAIX, with a low CAIX/CAII selectivity (*K*_i_ = 0.07 μM for CAII). The [^18^F]F-labeled cationic compound was evaluated in mice bearing HT-29 cancer xenografts, showing moderate tumor uptake of 0.41% ID/g at 1 h PI. 

### 4.2. Acetazolamide (AAZ)

AAZ is a clinically approved CA inhibitor for the treatment of glaucoma, heart failure, and altitude sickness. It is also found to be effective in treating cancers by blocking some of the enzymes needed for tumor cell growth. From the structural point of view, AAZ is a small, heteroaromatic sulfonamide, which binds to CAIX by coordinating a zinc ion within the enzyme active site. The binding affinity (*K*_d_) of AAZ toward CAIX is reported to be in the range of 10 to 50 nM [66]. The very first in vivo AAZ-based fluorescence imaging study was reported by Ahlskog et al. [43]. Compound **10** accumulation in a tumor was confirmed by fluorescence microscopy analysis of tissue sections from LS174T xenograft-bearing nude mice (Figure 4). The fluorescence signals in all organs, except tumors, substantially decreased at 2 h. However, fluorescein (FITC) is not an ideal fluorophore for non-invasive imaging because its wavelength has poor tissue penetration. Soon after, Groves et al. replaced the FITC fluorophore with a NIR tag (VivoTag-680, Ex/Em = 670/686 nm) to produce **11** [44]. Compound **11** exhibited nanomolar inhibition toward CAIX with a 33-fold selectivity over other CA isoforms. The FMT imaging of **11** in HT-29 xenografted mice demonstrated successful tumor delineation at 24 h post-IV injection. The authors also found that the fluorescence signal in the tumor could be upregulated by maintaining the tumor-bearing mice in a hypoxic (8%) atmosphere instead of normoxic conditions, confirming that the accumulation of the probe was caused by hypoxia [67]. Another example was given by Mahalingam et al., who combined a bright NIR dye S0456 (Ex/Em, 776/796 nm) and AAZ to produce HypoxyFluor-1 (**12**) [45]. **12** showed excellent binding affinity toward the CAIX-expressing HT-29 cell line, with a *K*_d_ value of 10 nM. As shown in Figure 5, in vivo accumulation of the probe in mouse xenografts was determined by optical fluorescence imaging and confirmed by biodistribution. Notably, tumor contrast remained prominent for at least 24 h, with some tumor fluorescence clearly visible at 48 h PI.

In addition to fluorescence imaging, AAZs have also been applied to generate radiotracers for CAIX detection. Building on the initial success of the fluorescein-labeled probe, Krall and colleagues developed the [^99m^Tc]Tc-PHC-102 (**13**), also known as “Onco IX”, using a tripeptide (Lys-Asp-Cys) chelator for ^99m^Tc labeling [46]. The probe showed a high tumor uptake and an excellent SPECT imaging contrast in the renal cell carcinoma SKRC-52 model, but also a high kidney uptake (Figure 6). In the biodistribution study, maximal tumor uptake of the radiotracer was observed at 3 h PI (22% ID/g), with a T/B ratio of 70. This compound also exhibited good retention in tumors with 19.8% ID/g at 6 h PI The success of the preclinical results prompted the authors to initiate a phase I clinical study with 5 patients with RCC to assess the feasibility and safety of [^99m^Tc]Tc-PHC-102. For the SPECT/CT scans, the patients received a dose of 50 μg/600–800 MBq of the radiotracer [68]. Among the 5 patients, 3 patients who had histology-confirmed CAIX-expressed RCC showed high and sustained uptake of [^99m^Tc]Tc-PHC-102 in the tumor, stomach, and kidney over 6 h. One unexpected pulmonary metastatic lesion of 2.3 cm was also detected. Weak tracer accumulation was found in the other two patients who had histology-CAIX negative. The overall effective dose per patient was 6.3 ± 1.7 mSv. In the study, [^99m^Tc]Tc-PHC-102 demonstrated the potential to identify primary and metastatic RCC lesions in patients.

AAZ was radiolabeled with position emitters, such as ^18^F and ^68^Ga, for PET imaging. More et al. reported the synthesis and evaluation of [^18^F]F-AAZ (**14**) [47]. The fluorine-18 was performed by coupling an [^18^F]F-PEG-alkyne with an azide-substituted AAZ via a copper(I)-catalyzed click reaction, giving a radiochemical yield of 32.5%. Unlike previous probes for RCC detection, in vivo studies of the radiotracer were performed in 4T1 mouse mammary carcinoma and HT-29 human colorectal carcinoma xenograft models. In both models, trace amounts of **14** were found in tumors, while it mostly accumulated in the intestine, kidney, and stomach. Furthermore, the PET scans showed very low and variable signal levels in the tumors over the selected time points (5 to 90 min). The authors concluded from the in vivo studies that **14** led to insufficient accumulation in CAIX-expressing tumors. However, the low specificity could also be caused by the in vivo instability of the probe. (Only 20% of the probe was intact after 1 h PI.) Recently, we took advantage of a CBT/1,2-aminothiol click reaction to radiolabel AAZ with ^68^Ga. The high efficiency of the click reaction allowed us to establish a library approach to generating various compounds for screening. In the series of compounds, a new NODAPy chelator was used to complex ^68^Ga. **15** was obtained with a radiolabeling yield of >95% in 15 min. The biological results of the compounds have not been reported yet [48].

### 4.3. Saccharin

Saccharin is an artificial sweetener that possesses high affinity for CAIX compared to CAII (>50-fold) [69]. An in vivo biodistribution comparison of NIR-labeled AAZ **12** and saccharin **16** was performed by Mahalingam et al. in 2018. Saccharin-based compound **16** had a similar, but slightly lower, accumulation in all organs compared to the AAZ derivative. However, the tumor uptake of **16** was significantly lower than that of **12** (Figure 7) [45]. Very recently, Shin et al. performed the synthesis and evaluation of [^68^Ga]Ga-NOTA-SAC (**17**) for CAIX-overexpressing U87MG tumor detection [49]. The ^68^Ga tracer remained in the U87MG tumor for at least 90 min with 1.2~1.5% ID/g. They observed that the tracer distribution in all organs decreased over time by PET imaging, and the highest T/M ratio (4.17-fold) was reached at 90 min PI. Compared to saccharin-free ^68^Ga or [^68^Ga]Ga-NOTA, [^68^Ga]Ga-NOTA-SAC allowed for the clear visualization of the U87MG tumor.

### 4.4. Imidazothiadiazole Sulfonamide (IS)

Imidazothiadiazole sulfonamide is a new CAIX ligand identified by Ono and coworkers (Figure 8). In 2020, they reported the synthesis and biological evaluation of the IS-based [^111^In]In-DO3A-IS1(**18**) [50]. Tracer **18** exhibited higher binding to CAIX-positive HT-29 cells (118 ± 21% initial dose/mg protein) than CAIX low-expressing MDA-MB-213 cells (1.4 ± 0.3% initial dose/mg protein), indicating it possesses good selectivity. SPECT/CT imaging and biodistribution studies of **18** were performed in mice bearing HT-29 cancer xenografts. High uptake of **18** in the HT-29 tumor (8.71% ID/g at 24 h PI) and rapid clearance from the blood pool were observed (T/B ratio = 14.4, 21.7, 29.1, and 53.6 at 1, 4, 8, and 24 h, respectively). The tumors were clearly visualized at 4 h, and the best imaging contrast was found at 24 h PI However, high accumulation in the kidney was observed (65.1–115% ID/g at 1–24 h PI). In order to reduce the uptake in non-target organs, they introduced an albumin binder (ALB) into **18** to generate [^111^In]In-DO2A-ALB1 (**19**) [51]. ALB moieties, such as 4-(*p*-iodophenyl)-butyric acid, exhibit reversible binding to albumin in the blood with a micromolar affinity, which prolongs the circulation time of the tracer, thereby enhancing its tumor uptake. The additional ALB moiety slightly compromised the binding affinity (IC_50_ values were 127 and 574 nM for **18** and **19**, respectively). Biodistribution and SPECT/CT imaging were performed in mice bearing HT-29 cancer xenografts. Tumor uptake of **19** was 12.32% ID/g at 48 h PI, which was significantly higher than the uptake of **18** (5.37% ID/g at 48 h PI). On the other hand, **19** had a significantly lower renal accumulation (17.55−56.06% ID/g) than **18** (91.30% ID/g), indicating that the introduction of ALB enhanced the blood circulation of the IS compound and concomitantly increased the tumor uptake. Coronal SPECT/CT images of an HT-29 tumor-bearing mouse obtained at 24 h PI after administration of **19** showed a high tumor-imaging contrast, confirming that [^111^In]In-DO2A-ALB1 may be an effective CAIX imaging probe (Figure 9).

### 4.5. Multivalent CAIX Ligands

Imaging probes with multivalent ligands can improve the sensitivity of tumor detection by enhancing the target-binding affinity and extending the tumor retention time (Figure 10). An IRDye750-labeled bivalent AAZ (**20**) was reported by Krall et al. [52]. In the SPR-binding affinity assay, the bivalent probe exhibited excellent binding affinity toward CAIX and did not dissociate from the enzyme, while its monovalent analog was rapidly dissociated. The in vivo results revealed that the best imaging contrast after administration of **20** was at 24 h, with a SKRC-52 tumor uptake of 5.3% ID/g. A comparison of **20** and its monovalent analog showed that the bivalent tracer had a longer residence time inside the tumor, indicating the potential of the multivalent strategy for imaging of CAIX-positive cancer. NIR dyes possessing brighter fluorescence emission and longer emission wavelengths can improve imaging resolution for in vivo, deeper tumor detection. In 2016, Lv et al. reported the synthesis and evaluation of the HypoxyFluor (**21**), containing a hydrophilic PEG linker and a bivalent BSA moiety [53]. The probe had a slightly lower binding affinity (*K*_d_ = 45 nM) compared to the unmodified ligand in an HT-29 cell-based binding assay. The compromised CAIX binding affinity may be caused by the attachment of the long PEG linker and the fluorophore. For the imaging study, **21** was administrated to HT-29 tumor-bearing mice. Fluorescence was restricted to the tumor and the kidney when low doses were administered, e.g., 3 and 13 nmol, but additional fluorescence was seen in the liver when animals were treated with a higher dose (40 nmol). Although the fluorescent signal was predominately detected in the tumor at 4 h PI, most of the fluorescence was lost from the tumor at 8 h PI.

The multivalent strategy has also been applied to nuclear imaging probes. For SPECT imaging, Lv et al. recently reported the synthesis of [^99m^Tc]Tc-labeled bivalent BSAs containing long PEG linkers [54]. Their report showed that the bivalent BSA **22** had a better binding affinity than its monovalent analog (57 nM vs. 146 nM). Compound **22** was administrated into mice bearing HT-29 xenografts for in vivo, preclinical evaluation. The tumor uptake of **22** was about 5% ID/g, and in the SPECT images, the bivalent tracer remained detectable in the tumor at 9 h PI. However, the low T/B ratio (<1) hampered the discrimination of the tumor from the background tissues.

The bivalent CAIX ligand, US2, was designed with two ureido-sulfonamides introduced in the opposite positions of a 1,4,7,10-tetraazacyclododecane-1,7-diacetic acid (DO2A) chelator. The DOTA-based US2 allowed the labeling with various radiometals, such as ^111^In, ^90^Y, ^68^Ga, and ^67^Ga, for imaging or radionuclide therapy. The small size and multivalence of US2 were expected to provide fast clearance from blood and high accumulation in the tumor. Iikuni et al. reported the synthesis and evaluation of a series of ^111^In- and ^90^Y-labeled ureido-sulfonamides, [^111^In]In-US2 (**23**) and [^90^Y]Y-US2 (**24**), for SPECT imaging and therapy [55]. In the in vitro cell-uptake assay, the authors observed that **23** showed significantly higher binding to CAIX-overexpressing cells than its monomeric analog ([^111^In]In-US1), implying that bivalency maximized the uptake of the probe. In vivo, **23** was found to accumulate selectively in HT-29 tumors (4.57% ID/g at 1 h PI) and was rapidly cleared from the blood pool and muscle after 4 h. Visualization of HT-29 tumors in mice at 1 h PI was not achieved with SPECT imaging because of the high blood background, but the tumor could be seen after 4 h PI. In addition, they demonstrated that using **24** for radiotherapy could significantly delay HT-29 tumor growth, compared to that in untreated mice, without any critical hematological toxicity. The US2 moiety was labeled with ^68^Ga by Iikuni et al. to provide **25** for PET imaging [56]. The binding of Ga-labeled US2 to HT-29 cells was demonstrated by using [^67^Ga]Ga-US2, showing a value of 60.4% initial dose/mg protein. In the blocking study, the addition of AAZ significantly blocked the binding of US2 to the CAIX (10.8% initial dose/mg protein), confirming the in vitro CA-specificity. Biodistribution was performed using the longer half-life [^67^Ga]Ga-US2 in an HT-29 tumor-bearing mouse model. [^67^Ga]Ga-US2 accumulated in the HT-29 tumor (4.63 and 3.81% ID/g at 0.5 and 1 h PI, respectively). The T/B and T/M ratios were above 1.5 at 0.5 h PI. In the PET/CT images, the HT-29 tumors were clearly visualized at 1 h PI. Compared to **23**, **25** exhibited slightly lower uptake in HT-29 tumors, but a faster clearance from the blood and the muscle. The difference between the two US2-based imaging agents could come from the difference in their hydrophilicity and metal-complexing geometry. Inspired by the previous success of the IS-based **18**, Nakashima et al. investigated the effects of the replacement of the ureido-sulfonamide of **25** by the IS. [^67/68^Ga]Ga-DO2A-IS2 (**26**) and its monomeric analog [^67/68^Ga]Ga-DO3A-IS1 were prepared and tested in HT-29 tumor-bearing mice [57]. In the cell-binding assay, dimeric **26** showed significantly stronger binding to HT-29 cells (859 ± 71.7% initial dose/mg protein) than its monomeric analog (77.4 ± 4.8% initial dose/mg protein). However, although **26** exhibited better in vitro binding affinity than [^67^Ga]Ga-DO3A-IS1, the biodistribution of **26** revealed a lower uptake in HT-29 tumors (0.71 ± 0.06% ID/g at 1 h PI) compared to the monomeric analog (1.92 ± 0.16% ID/g at 1 h PI). Moreover, high accumulations of both agents in normal organs, such as the pancreas, intestine, kidney, liver, heart, and lungs, were observed. As a result, the HT-29 tumor could not be visualized by using the Ga-labeled US2 in the SPECT images. 

A hydroxamamide (Ham) chelator provides a [^99m^Tc]Tc complex consisting of two Ham ligands, which is ideal for the preparation of bivalent targeting probes. Iikuni et al. synthesized and evaluated a Ham-based [^99m^Tc]Tc-URB2A (**27**) [58]. The radiolabeling yield of this probe was 39.1%, and the radiochemical purity was higher than 95%. The uptake of **27** into HT-29 tumors was 3.44 ± 0.50% ID/g at 1 h PI. Although the uptake was markedly greater than other previous reports, the T/B ratio remained insufficient for in vivo imaging. As a result, the authors further developed [^99m^Tc]Tc-ISB2 (**28**), containing two Ham-IS moieties [59]. The radiosynthesis was performed by mixing [^99m^Tc]Tc-pertechnetate, the Ham-IS precursor, and tin(II) tartrate hydrate to give the desired **28** in 67% radiochemical yield. The binding affinity of **28** toward HT-29 cells was determined to be 703 ± 31% initial dose/mg protein, which is 7.5-fold greater than that of **27**. However, **28** was found to be less stable (55% intact compound after 6 h incubation) than **27** (80% after 7 h incubation). The HT-29 tumor uptake of **28** was 1.82% ID/g at 1 h PI, which was lower than that of **27** (3.44% ID/g at 1 h PI). The lower tumor accumulation might be due to its poor stability. **28** exhibited lower blood retention compared to **27**, but the low T/B ratio was unsatisfying for imaging. 

Considering that a multivalent targeting strategy can potentially improve retention and affinity, Lau et al. developed a trimeric-[^18^F]F-sulfonamide **29** [41]. The trimeric probe exhibited good affinity toward CAIX (*K*_i_ = 8.5 nM). However, it was not superior to the corresponding monomeric **8**, and a loss of CAIX/CAII selectivity was reported. Tumor uptake of **29** (0.33% ID/g at 1 h PI) was lower than the uptake of the corresponding monomer, but the tumor-to-background ratios were significantly improved by the trimeric strategy (e.g., T/M ratios of **8** and **29** were 2.15 and 9.55, respectively). Therefore, clear visualization of the CAIX–expressing HT-29 tumor xenografts from PET scans was possible with **29**. Based on this multivalent strategy, a systematic study of monomeric [^68^Ga]Ga-DOTA-AEBSA (**30**), dimeric [^68^Ga]Ga-DOTA-(AEBSA)_2_, (**31**) and trimeric [^68^Ga]Ga-NOTGA-(AEBSA)_3_ (**32**) for PET imaging in mice bearing an HT-29-human-colorectal-carcinoma xenograft was performed by the same group [60]. The binding affinity for CAIX was determined by using their corresponding non-radioactive analogs, giving *K*_i_ values between 7 and 10 nM. However, the CAIX/CAII selectivity was not improved by the multivalency. Tumor uptake of these compounds ranged from 0.81 to 2.30% ID/g at 1 h PI, whereas T/M ratios varied between 4.07 and 5.02. Despite the low contrast, accumulation of the three probes was enough to visualize the tumors. 

### 4.6. Dual-Motif CAIX Inhibitor: XYIMSR

A multivalent strategy using heterologous ligands is also attractive for developing CAIX-targeted imaging probes (Figure 11). Wichert et al. reported a dual-motif CAIX inhibitor, XYIMSR, containing an AAZ and a biphenolic moiety via the dual-display DNA-Encoded Libraries (DEL) approach [70]. The XYIMSR showed an excellent binding affinity toward CAIX, with a *K*_d_ value of 0.2 nM. In their study, the IRDye750-labeled XYIMSR showed long residence time and high tumor uptake (10% ID/g) in mice bearing SKRC-52 xenografts at 24 h PI However, similarly high uptake (~9% ID/g) by the kidney was observed at the same time point. In 2015, Yang et al. radiolabeled XYIMSR with ^111^In to generate [^111^In]In-XYIMSR-01 (**33**) for the imaging of ccRCC [61]. The IC_50_ of the [^113/115^In]XYIMSR-01 was 108.2 nM, while the FITC-labeled analog was 0.2 nM. Tumor-selective uptake and retention of the radiotracer were confirmed by biodistribution and SPECT/CT imaging. At 1 h post-injection, 26.0% ID/g of radiotracer uptake was observed within the tumor. The maximal uptake was at 8 h PI. with 34.0% ID/g, and the tumor-to-kidney ratio was 3.1. SPECT/CT imaging revealed radiotracer uptake in the tumor as early as 1 h PI. and retention up to 48 h. Then, dosimetry, toxicity, and chemistry manufacturing and controls (CMC) studies were performed to translate **33** into phase I clinical trials [71]. Moreover, XYIMSR was radiolabeled with ^64^Cu to generate [^64^Cu]Cu-XYIMSR-06 (**34**) for the PET imaging of ccRCC [72]. The maximal tumor uptake was observed at 4 h PI (19.3% ID/g), which was lower than that for the uptake of the ^111^In-labeled analog. On the contrary, **34** showed better imaging contrast because the ratio of tumor-to-normal tissues was improved (e.g., the tumor-to-kidney ratio was reduced to 1.0). 

In addition to ccRCC imaging, **34** was applied to the PET imaging of malignant glioma in U87 MG tumor cell xenograft mice [62]. The binding affinity of **34** toward CAIX in U87 MG glioma was 4.22 nM (*K*_d_ value). Maximal tumor uptake of **34** was observed at 4 h with 3.13% ID/g. The clearance from the tumor was slow, down to 0.81% ID/g by 24 h PI. However, high uptake of the radiotracer in the kidney, stomach, lung, and intestines was observed. MicroPET imaging of orthotopic gliomas revealed that the best imaging contrast in tumors was obtained at 8 h, while the tumor-to-brain and tumor-to-muscle ratios were 5.4 and 5.3, respectively (Figure 12). The authors suggested that **34** has potential for further clinical applications. Recently, a new ^111^In-labeled dual-CAIX-targeted probe containing a CA9tp peptide (NHVPLSP) and an AAZ moiety was reported [73]. AAZ-CA9tp showed high selectivity toward CAIX (*K*_d_ = 8.7 nM toward rhCA9; *K*_d_ = 226.4 nM toward rhCA2) and possessed good CA9-targeting ability in hypoxic HCT15 cells (*K*_d_ = 6.17 nM). In their in vivo study, the [^111^In]In-DOTA-AAZ-CA9tp had a significant tumor uptake (>25% ID/g) and remarkable T/B (>150) and T/M (>50) ratios at 24 PI in HCT15 tumor-bearing mice. The combination of small-molecular- and peptide-based CAIX ligands may provide advantages to the pharmacokinetic properties of the final conjugates.

Based on the XYIMSR, Huang et al. recently developed \CAIX-specific probe **35** with an IRDye 800CW moiety (CAIX-800) for a multimodality imaging method of a combination of fluorescence molecular tomography-computed tomography (FMT-CT) and multispectral optoacoustic tomography (MSOT) [63]. FMT allows the tracking of the fluorescent probe in vivo, whereas CT enables precise 3D localization of the hypoxia biomarker CAIX with high sensitivity. MSOT takes advantage of the photoacoustic effect to generate high-resolution optical imaging in scattering media, such as the human body. In MSOT, the probe could be used as an exogenous contrast agent. The probe for FMT-CT and MSOT imaging was evaluated in an orthotopic nasopharyngeal carcinoma (NPC)-bearing mouse model. FMT-CT exhibited a better NPC contrast than MSOT in the early stage of NPC, probably because the IRDye 800 is primarily designed for fluorescence imaging (normally more sensitive than optoacoustic imaging). However, the FMT-CT could not distinguish between the primary and lymph node metastases at the advanced stage of orthotopic NPC. On the contrary, the high-resolution MSOT allowed superior visualization of the primary and metastatic tumors. In order to make this type of dual-modality probe more successful, the development of a new molecule with a good balance between FMT fluorophore and MSOT exogenous contrast agent is suggested. 

## 5. Conclusions

CAIX is a cellular-membrane-bound protein expressed in tumor hypoxia and certain malignancies, such as ccRCC. Clinical evidence showed that the aberrant expression of CAIX is associated with poor prognosis and therapy outcome, making it a relevant target for molecular imaging, intraoperative detection, and follow-up after treatment. Among the numerous CAIX-targeted imaging probes reported, radiolabeled girentuximab-based tracers are the most clinically investigated agents, and it was recently demonstrated that [^89^Zr]Zr-labeled girentuximab could successfully distinguish ccRCC from non-ccRCC lesions. However, the slow uptake and clearance of antibodies requires a long time (usually 3 to 7 days) to achieve optimal imaging. As a result, the radiation burden on the patient is relatively high in comparison to small-molecular tracers that are cleared more rapidly from the body. This could be a limitation for the future development of CAIX-targeted, antibody-based imaging agents. The accurate assessment of radiation and protein doses may be the next key step in the successful deployment of mAb-based imaging probes. The advanced application of mAb carriers will likely be as dual-modality imaging probes to help doctors in the preoperative assessment and intraoperative delineation of tumor margins. This could be a powerful tool for precise surgery as long as the nuclear and fluorescence imaging devices are well-integrated into the surgical practice.

It is expected that low-molecular-weight biovectors will continue to play a central role in the design and development of CAIX-targeted imaging probes. Compared full-length antibodies, imaging probes with lower molecular weights possess fast pharmacokinetics, allowing patients to complete imaging examinations in one day. A wide variety of CAIX-targeted ligands (e.g., aromatic sulfonamides, peptides, antibody fragments, and affibodies) have been conjugated with various imaging modalities for optical, PET, SPECT, and multimodality imaging. The current selection of the lead probes is mainly based on their in vitro binding affinity. However, under many circumstances, the in vitro binding assay results are not positively correlated to the in vivo tumor uptake. This might be due to the difference in CAIX-expression levels in the tumor models. As ccRCC always expresses high-level CAIX, many studies focus on the development of probes to image ccRCC. However, although most probes showed excellent uptake in ccRCC tumors, non-specific accumulation in kidneys is often too high. This could be a challenge for the detection of primary ccRCC in patients. 

For small-molecular probes, modifications of the pharmacophore to improve recognition by CAIX-expressing cells is an important strategy for increasing tumor uptake and imaging contrast. Many multivalent and dual-motif CAIX ligands have been prepared and exhibited better binding affinity, longer tumor retention time, and higher tumor uptake than their monovalent analogs. On the other hand, emerging studies on the development of CAIX-specific small-molecules (lower binding affinity toward other CAs) have gained great attention. A current trend is to target the PG-like domain of CAIX. Since the PG-like domain is unique in CAIX, the development of imaging probes targeting the PG-like domain of CAIX should be emphasized. Another attractive strategy is to elaborate molecules targeting the area outside the active site of the enzymes where CAIX is different from other CAs. Dudutiene et al. reported a new class of CAIX inhibitors based on fluorinated BSA, and the lead compound had an excellent selectivity between CAII and CAIX (CAII/CAIX = 1300) [74]. Investigation of new, highly selective CAIX inhibitors may shed light on a new research path for CAIX imaging. 

CAIX-targeted imaging holds a bright future for the accurate treatment planning for individual patients, as many types of solid tumors share the common feature of CAIX-overexpression. Many probes have advanced to clinical trials and are safe. In future efforts towards developing better CAIX-targeted imaging probes, cooperative efforts are needed from chemists, biologists, and clinicians to overcome the current limitations.

## Figures and Tables

**Figure 1 ijms-23-06125-f001:**
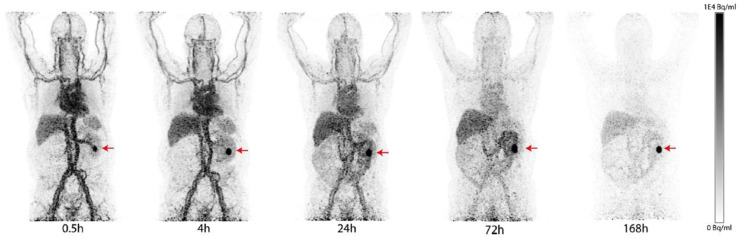
PET/CT imaging of a patient with a ccRCC tumor in the left kidney (red arrow) after injection of [^89^Zr]Zr-girentuximab. The patient received a mass dose of 10 mg of girentuximab and was imaged at 0.5 to 168 h PA. Tumor-to-background ratio was increased over time.

**Figure 2 ijms-23-06125-f002:**
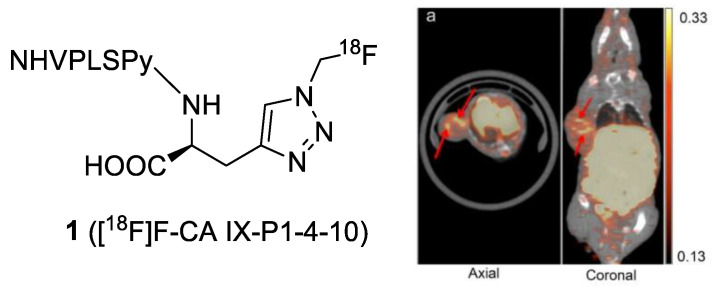
MicroPET/CT images acquired at 1 h PI of [^18^F]F-CaIX-P1-4-10 in HT-29 colorectal cancer xenograft-bearing mice. Arrows indicate high-count density regions in the tumor. The units of the bar scale are standard uptake values (SUVs).

**Figure 3 ijms-23-06125-f003:**
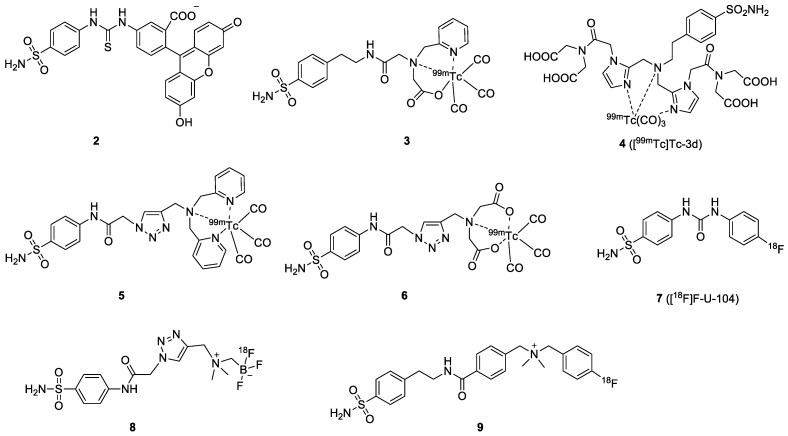
Chemical structures of BSA-based CAIX probes.

**Figure 4 ijms-23-06125-f004:**
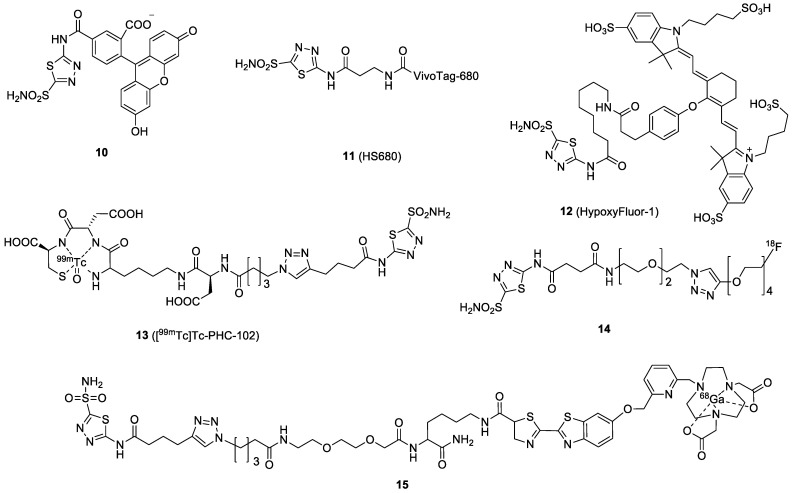
Chemical structures of AAZ-based CAIX probes.

**Figure 5 ijms-23-06125-f005:**
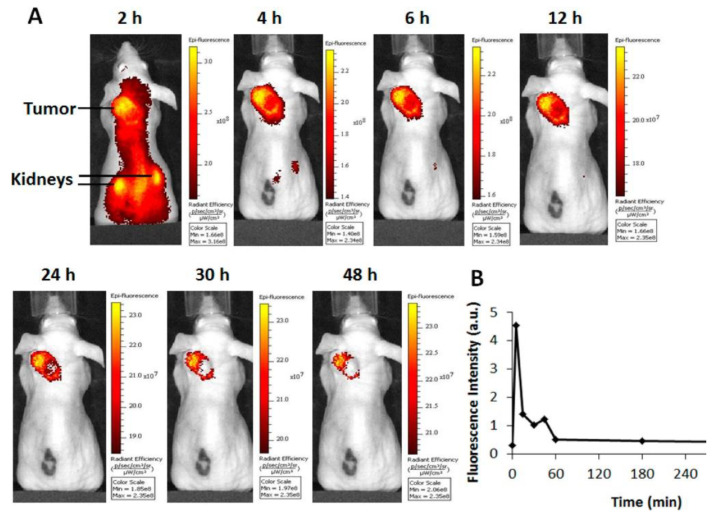
(**A**) Tumor accumulation and circulation half-life of HypoxyFluor-1 (**12**) in CAIX positive HT-29 tumor-bearing mice. (**B**) The tracer was eliminated from blood within 1 h.

**Figure 6 ijms-23-06125-f006:**
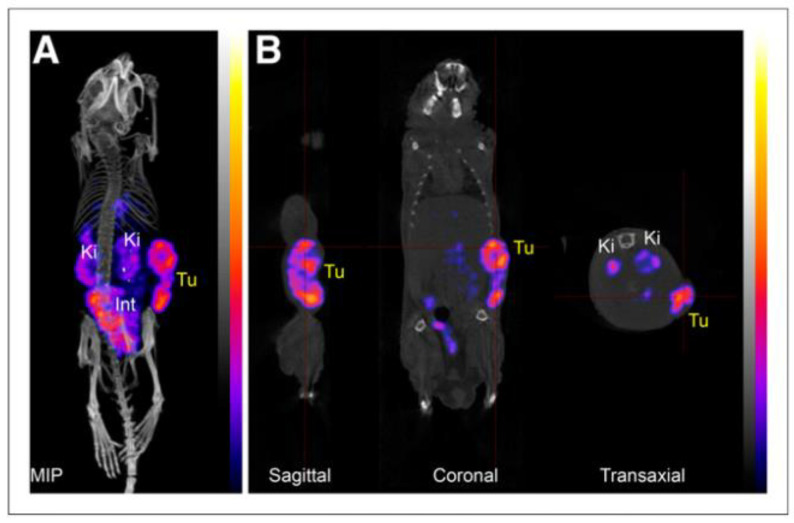
SPECT/CT scan with [^99m^Tc]Tc-PHC-102 (**13**) in SKRC-52-bearing mouse at 4 h PI (**A**) Maximum-intensity projection (MIP) and (**B**) sagittal, coronal, and trans-axial projections. Int = intestine; Ki = kidney; Tu = tumor.

**Figure 7 ijms-23-06125-f007:**
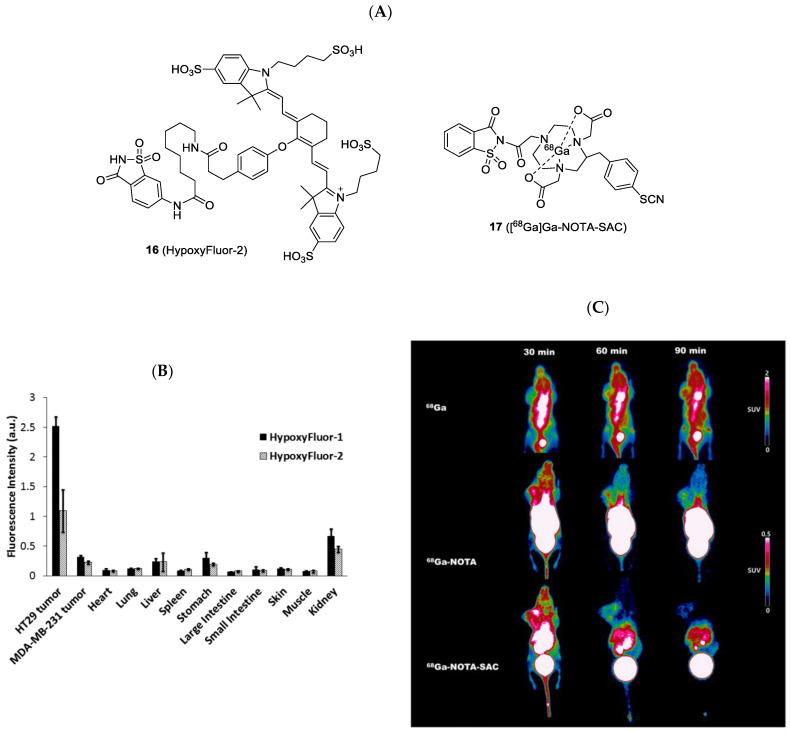
(**A**) Chemical structure of **16** and **17**. (**B**) Comparison of the fluorescence intensity of HypoxyFluor-1 (**12**) and HypoxyFluor-2 (**16**) in HT-29 tumor-bearing mice. (**C**) PET image analysis of free ^68^Ga, [^68^Ga]Ga-NOTA, and [^68^Ga]Ga-NOTA-SAC **17** at 30, 60, 90, min PI, respectively.

**Figure 8 ijms-23-06125-f008:**
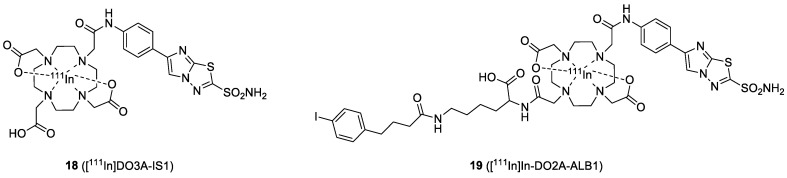
Chemical structures of IS-based CAIX imaging probes.

**Figure 9 ijms-23-06125-f009:**
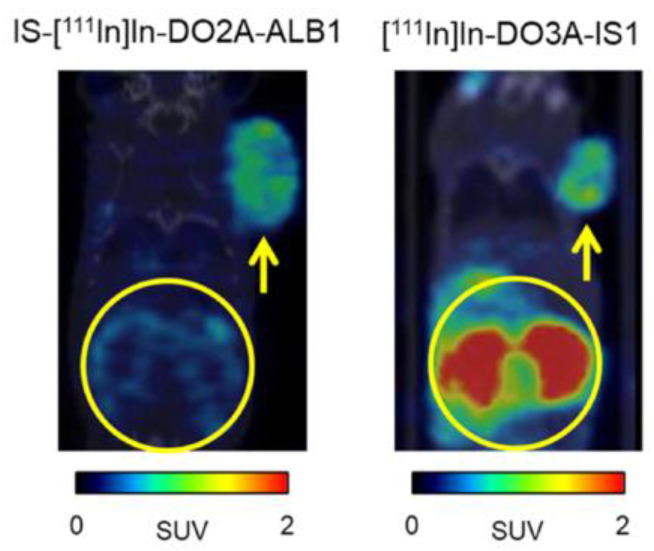
Coronal SPECT/CT images of an HT-29 tumor-bearing mouse obtained at 24 h PI using IS-[^111^In]In-DO2A-ALB1 and [^111^In]In-DO3A-IS1.

**Figure 10 ijms-23-06125-f010:**
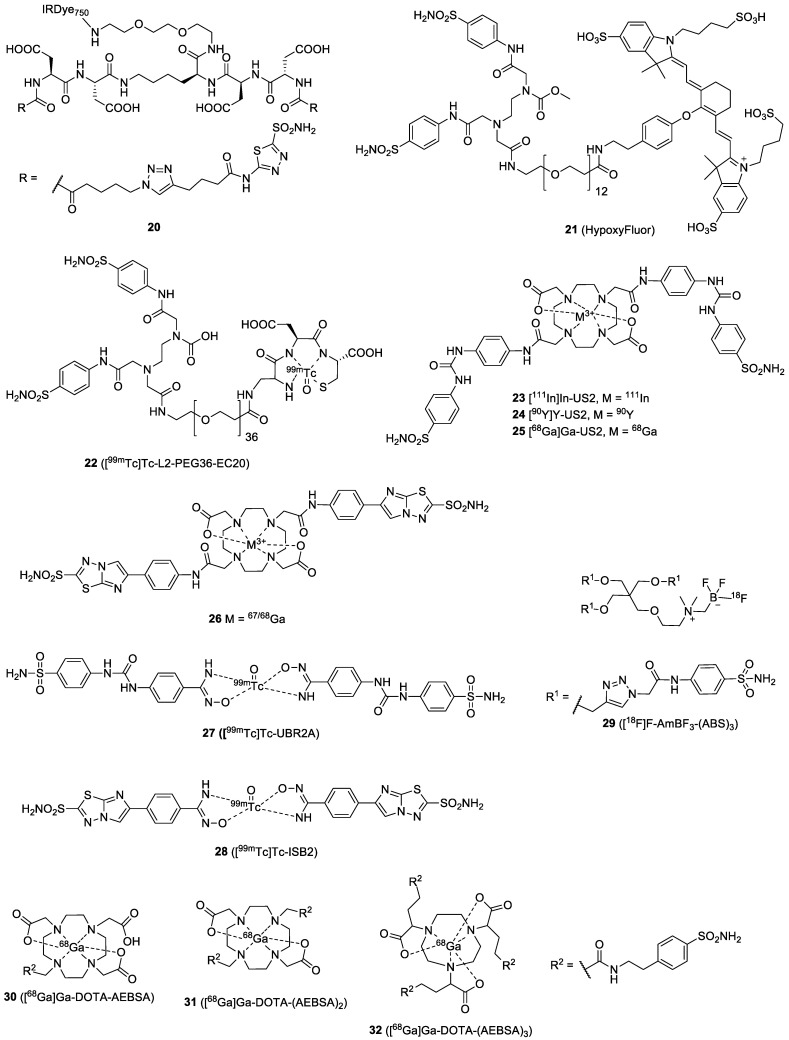
Chemical structures of published multivalent CAIX ligands.

**Figure 11 ijms-23-06125-f011:**
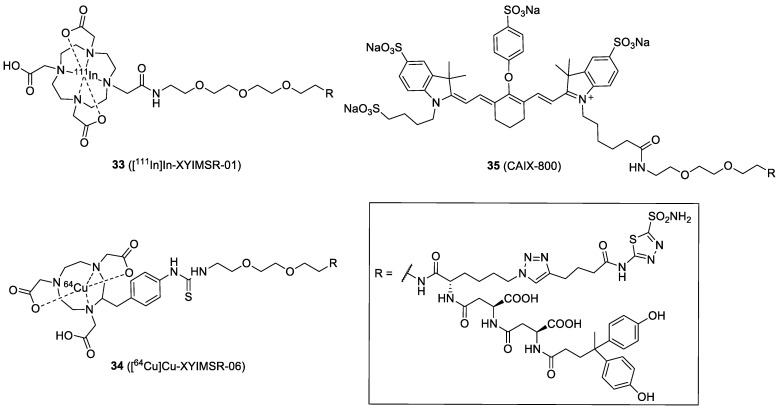
Chemical structures of the dual-motif CAIX-targeted probes.

**Figure 12 ijms-23-06125-f012:**
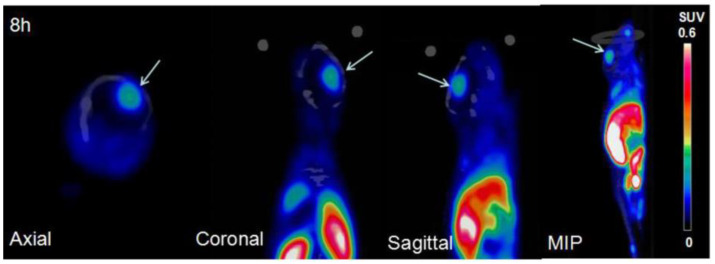
Micro-PET images of orthotopic U87MG xenografts of [^64^Cu]Cu-XYIMSR-06 (**34**) at 8 h PI.

**Table 1 ijms-23-06125-t001:** Overview of the preclinical evaluation of radiolabeled, CAIX-targeted mAb and its fragments.

Compound	Tracer Type	Tumor Model	Tumor Uptake (% ID/g)	T/B *^a^*	T/M *^b^*	Time p.i.	Year
[^125^I]I-girentuximab-IRDye800CW	antibody	SKRC-52	6.9	- *^c^*	- *^c^*	72 h	2014 [15]
[^111^In]In-DTPA-G250-IRDye800CW	antibody	SKRC-52	58.5	- *^c^*	- *^c^*	168 h	2015 [16]
[^89^Zr]Zr-DFO-cG250-F(ab’)_2_	F(ab’)_2_	SCCNij-3	3.71	0.9	6.8	4 h	2010 [18]
[^111^In]In-DTPA-cG250-F(ab’)_2_	F(ab’)_2_	SCCNij-153	4.1	30.8	12.1	24 h	2017 [19]
[^111^In]In-DTPA-cG250-F(ab’)_2_	F(ab’)_2_	FaDu	5.61 at 24 h PI (before treatment) 1.92 at 24 h PI (after treatment)	- *^c^*	- *^c^*	24 h	2021 [21]
[^111^In]In-DTPA-VHH-B9	nanobody	SCCNij-153	0.51	8.1	19.6	4 h	2022 [22]
[^111^In]In-DOTA-ZCAIX:2	affibody	SCCNij-153	0.32	4.9	3.3	4 h	2019 [23]
[^111^In]In-DOTA-ZCAIX:2	affibody	SKRC-52	15	63	102	4 h	2019 [24]

*^a^* T/B = tumor-to-blood ratio. *^b^* T/M = tumor-to-muscle ratio. *^c^* “-“ = not reported in the paper.

**Table 2 ijms-23-06125-t002:** Overview of preclinical evaluation of small-molecule-based CAIX-targeted probes.

Agent	Imaging Modality	Fluorophores/Radioisotopes	Binding Affinity	Selectivity CAII/CAIX	Model	Tumor Uptake	Tumor Visualization	Ref.
**1**	PET	^18^F	N/A	N/A	HT-29	SUVmean = 0.38 ± 0.03 at 1 h	Y	[31]
**2**	FL	FITC	*K*_i_ = 24 nM	NA	HT-29	N/A	Y	[36]
**3**	SPECT	^99m^Tc	*K*_i_ = 58 nM	0.86	HT-29	0.2 ± 0.1% ID/g at 0.5 h	Y	[37]
**4**	SPECT	^99m^Tc	IC_50_ = 9 nM	N/A	HeLa	N/A	N/A	[38]
**5**	SPECT	^99m^Tc	*K*_i_ = 0.22 µM *^a^*	0.3	HT-29	0.07 ± 0.03% ID/g at 1 h	N/A	[39]
**6**	SPECT	^99m^Tc	*K*_i_ = 0.037 µM *^b^*	1.2	HT-29	0.14 ± 0.10% ID/g at 1h	Y	[39]
**7**	PET	^18^F	*K*_i_ = 45 nM	2.1	HT-29	0.83 ± 0.06% ID/g at 1 h	Y	[40]
**8**	PET	^18^F	*K*_i_ = 6.6 nM	9.0	HT-29	0.64 ± 0.08% ID/g at 1 h	Y	[41]
**9**	PET	^18^F	*K*_i_ = 0.22 µM	0.3	HT-29	0.41 ± 0.06% ID/g at 1 h	Y	[42]
**10**	FL	FITC	N/A	N/A	N/A	N/A	N/A	[43]
**11**	FL	VivoTag-680	*K*_i_ = 7.5 nM	33.1	HT-29	10% ID/g at 24 h	Y	[44]
**12**	FL	S0456	*K*_d_∼10 nM.	N/A	HT-29	FL intensity: 2.5 (a.u.) at 4 h	Y	[45]
**13**	SPECT	^99m^Tc	N/A	N/A	SKRC-52	22% IA/g at 3 h	Y	[46]
**14**	PET	^18^F	N/A	N/A	4T1 HT-29	4T1: 0.2% ID/g at 30 min HT-29: 0.2% ID/g at 15 min	Y	[47]
**15**	PET	^68^Ga	N/A	N/A	N/A	N/A	N/A	[48]
**16**	FL	S0456	N/A	N/A	HT-29	FL intensity: 1.1 (a.u.) at 4 h	Y	[45]
**17**	PET	^68^Ga	N/A	N/A	U87MG	1.5% ID/g at 30 min	Y	[49]
**18**	SPECT	^111^In	118 ± 21% initial dose/mg protein	N/A	HT-29	8.71 ± 1.41% ID/g at 24 h	Y	[50]
**19**	SPECT	^111^In	IC_50_ = 574 nM	N/A	HT-29	12.32% ID/g at 48 h	Y	[51]
**20**	FL	IRDye750	k_a2_ = 1.36×10^−6^ RU^−1^ s^−1^	N/A	SKRC-52	5.3 ± 0.6% ID/g at 24 h	Y	[52]
**21**	FL	S0456	*K*_d_ = 45 nM	N/A	HT-29	Epi-fluorescence: 4.3 × 10^7^ at 4 h	Y	[53]
**22**	SPECT	^99m^Tc	*K*_i_ = 57 nM	N/A	HT-29	5% ID/g at 4 h	Y	[54]
**23**	SPECT	^111^In	125% initial dose/mg	N/A	HT-29	4.57 ± 0.21% ID/g at 1 h	Y	[55]
**25**	PET	^68^Ga	60.4 % initial dose/mg	N/A	HT-29	3.81% ID/g at 1 h	Y	[56]
**26**	PET	^68^Ga	859 ± 71.7 % initial dose/mg	N/A	HT-29	0.71 ± 0.06% ID/g at 1 h	N	[57]
**27**	SPECT	^99m^Tc	IC_50_ = 38.2 nM	N/A	HT-29	3.44 ± 0.50% ID/g at 1 h	N	[58]
**28**	SPECT	^99m^Tc	IC_50_ = 211.6 nM	N/A	HT-29	1.82% ID/g at 1 h	N/A	[59]
**29**	PET	^18^F	*K*_i_ = 8.5 nM	1.0	HT-29	0.33 ± 0.07% ID/g at 1 h	Y	[41]
**30**	PET	^68^Ga	*K*_i_ = 10.8 nM	12.7	HT-29	0.81 ± 0.15% ID/g at 1 h	Y	[60]
**31**	PET	^68^Ga	*K*_i_ = 25.4 nM	1.6	HT-29	1.93 ± 0.26% ID/g at 1 h	Y	[60]
**32**	PET	^68^Ga	*K*_i_ = 7.7 nM	0.93	HT-29	2.30 ± 0.53% ID/g at 1 h	Y	[60]
**33**	SPECT	^111^In	IC_50_ = 108.2 nM	N/A	SKRC-52	34.00 ± 15.16% ID/g at 8 h	Y	[61]
**34**	PET	^64^Cu	*K*_d_ = 4.22 nM	N/A	U87 MG	3.13 ± 0.26% ID/g at 4 h	Y	[62]
**35**	FMT-CT/MSOT	IRDye 800CW	N/A	N/A	NPC	N/A	Y	[63]

*^a^* N/A means not available. *^b^* The *K*_i_ was determined using the ^nat^Re-labeled analog.

## Data Availability

Not applicable.

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
