# Peer review of "New Developments in Carbonic Anhydrase IX-Targeted Fluorescence and Nuclear Imaging Agents"

_ijms, 2022, doi:10.3390/ijms23116125_

Round 1

Reviewer 1 Report

The review of K-T Chen and Y Seimbille describes the recent development of antibodies, peptides and small molecule CA IX inhibitors conjugated to fluorescence dyes or substituted with radioisotopes for imaging of solid tumor overexpressing CA IX in hypoxia conditions.

The review is well written, well structured and the figures and tables are helpful. I just have minor suggestions:

  • In the introduction I would change the following sentence “The overexpression of CAIX is mediated by transcription factors, such as the hypoxia- inducible factor 1-α (HIF1-α), which contribute to cancer cell ...” to avoid misunderstanding. In fact, it is not clear if the pronoun which refers to CAIX or the transcription factor. Also, I would spend more words on the CA family and include a recent review.

  • Pag. 6: CaIX-PI should be corrected to CaIX-P1.

  • Please explain in the text T/B ratio the first time you use it.

  • At line 300 please change “On contrary” with “On the contrary”.

  • Please explain the meaning of SUVmean.

  • The following sentence it is not readily understandable“The preference in binding of saccharin to CAIX over other CAs might be due to the fact that the entry of saccharin into the active site is similar to that of the substrate CO2”. Please rephrase it since it does not report clearly the results of the cited article.

  • Please use the name of the first author's name in all the in-text citations.

Author Response

We would like to thank the reviewer for the thoughtful comments on our manuscript. They are very helpful to improve the quality of our review paper. The point-by-point answer to the comments can be found below.

  1. In the introduction, I would change the following sentence “The overexpression of CAIX is mediated by transcription factors, such as the hypoxia- inducible factor 1-α (HIF1-α), which contribute to cancer cell ...” to avoid misunderstanding. In fact, it is not clear if the pronoun which refers to CAIX or the transcription factor. Also, I would spend more words on the CA family and include a recent review.

Few sentences were rewrote (main text, line 36-39) and two references (ref. 1 and 2) were added to describe CAIX more clearly.

  1. Page 6: CaIX-PI should be corrected to CaIX-P1.

We corrected the name of the compound in the manuscript.

  1. Please explain in the text T/B ratio the first time you use it.

It has been explained (line 163) to what T/B is corresponding.

  1. At line 300 please change “On contrary” with “On the contrary”.

It has been corrected according to the reviewer's suggestion.

  1. Please explain the meaning of SUV .

The meaning of SUV has been added at line 258.

  1. The following sentence it is not readily understandable “The preference in binding of saccharin to CAIX over other CAs might be due to the fact that the entry of saccharin into the active site is similar to that of the substrate CO2”. Please rephrase it since it does not report clearly the results of the cited article.

We agree with the reviewer and for clarity reason we decided to remove this sentence from the main text.

  1. Please use the name of the first author's name in all the in-text citations.

Thanks to the reviewer's recommendation, we changed the in-text citations.

Hopefully, the reviewer will find that our answers to the comments are satisfying.

Reviewer 2 Report

The review paper "New Development in Carbonic Anhydrase IX-Targeted Fluorescence and Nuclear Imaging Agents", written by Kuo-Ting Chen and Yann Seimbille, describes CAIX-targeted imaging agents and highlights strategies for enhancing imaging contrast with labeled antibodies, peptides, and small molecule-based probes. 

The paper can be published in the International Journal of Molecular Sciences after the revision of some minor errors and corrections. This will improve the quality and scientific rigor of the final version of the manuscript, in order for it to be readable to the International Journal of Molecular Sciences community. In terms of scientific content, the paper is well structured and organized, with no need for improvements in terms of subjects.

Following is a list of minor errors and corrections that I suggest be made in order to improve the quality of the paper to a final version.

Affiliation number 3 does not corresponds to any author; please correct.

Line 37: contributes instead of "contribute".

In lines 72 and 73 the authors refer to clinical trials of radio-labeled CAIX-targeted antibodies. It would be great if the authors could present in more detail some of the clinical trials already in the clinic and in preclinical stage.

Line 93: "girentumixab" was already described earlier in the text, it can be deleted here.

Line 155: define "p. a."

Line 127: define p. i. (post-injection).

Lines 169, 173, 182, 199: References' numbers are superscript and should be like the other ones.

Line 177: fail instead of "fails".

Line 191: Reference of the "comparison study" is missing; please indicate it.

Table 1. It should be written p. i. instead of "p.i"

Line 233: HCT116 instead of "HCT-116".

Figure 2. Indicate the units of the numbers in the color bar.

Line 292: "tumor-to-background" can be deleted because it was already described earlier.

Line 320: It is not well understood if 9-folds is for higher or lower. Indicate that, for example, "9-fold higher".

Line 340: The authors should indicate (FITC) after "fluorescein". 

Line 351: Figure 5 instead of "figure 5".

Line 364: 99mTC instead of "99mTC".

Line 390: "with" can be deleted.

Figure 7B - should be indicated in the figure which one is 12 (NIR-AAZ) and which one is 16 (NIR-saccharin).

Line 420: HT-29 instead of "HT29".

Lines 513 and 526: "imidazothiadiazole sulfonamides" and "hydroxamamide" can be deleted because they are already described earlier.

Legends of Figure 11 and Figure 12 should end with ".".

Table 2. Model instead of "model".

Author Response

We would like to thank the reviewer for the thoughtful comments on our manuscript. They are very helpful to improve the quality of our review paper. The point-by-point answer to the comments can be found below.

1. Affiliation number 3 does not corresponds to any author; please correct.

Thanks for noticing this. It corresponds to one of the affiliation of Dr. Yann Seimbille. Number 3 is now assigned.

2. Line 37: contributes instead of "contribute".

It has been changed according to the reviewer's suggestion. Note that line 36 to 39 were changed to describe CAIX better.

3. In lines 72 and 73 the authors refer to clinical trials of radio-labeled CAIX-targeted antibodies. It would be great if the authors could present in more detail some of the clinical trials already in the clinic and in preclinical stage.

We would like to thank the reviewer for this comment. Some information about the clinical or preclinical trials of several radio-labeled CAIX-targeted antibodies (e.g. [131I]I-labeled G250, [89Zr]Zr-girentuximab) is already included line 78-120 in our original manuscript. To keep the manuscript concise and to focus on the variety of imaging probes for CAIX, we decided to not add more details about the clinical trials.

4. Line 93: "girentumixab" was already described earlier in the text, it can be deleted here.

It was changed accordingly.

5. Line 155: define "p. a."

The definition of “p.a.” has been added in line 115.

6. Line 127: define p. i. (post-injection).

We added the definition of “p.i.” (line 127).

7. Lines 169, 173, 182, 199: References' numbers are superscript and should be like the other ones.

Thank you for the suggestion. It has been changed.

8. Line 177: fail instead of "fails".

It has been corrected.

9. Line 191: Reference of the "comparison study" is missing; please indicate it.

This comparison study was originally described in reference 23. We modified one sentence in line 192 in order to indicate the citation more clearly.

10. Table 1. It should be written p. i. instead of "p.i"

We changed it.

11. Line 233: HCT116 instead of "HCT-116".

We corrected this.

12. Figure 2. Indicate the units of the numbers in the color bar.

The figure was taken from the corresponding reference, but we described the unit of the bar scale in the figure caption.

13. Line 292: "tumor-to-background" can be deleted because it was already described earlier.

We deleted "(T/B)".

14. Line 320: It is not well understood if 9-folds is for higher or lower. Indicate that, for example, "9-fold higher".

It was changed according to the reviewer’s suggestion.

15. Line 340: The authors should indicate (FITC) after "fluorescein".

We changed it.

16. Line 351: Figure 5 instead of "figure 5".

We corrected it.

17. Line 364: 99m TC instead of "99mTC".

"99m" is now in superscript.

18. Line 390: "with" can be deleted.

"with" was deleted.

19. Figure 7B - should be indicated in the figure which one is 12 (NIR-AAZ) and which one is 16 (NIR-saccharin).

Thanks to the reviewer’s suggestion. We modified the names of the two compounds in Figure 7.

20. Line 420: HT-29 instead of "HT29".

It was changed accordingly.

21. Lines 513 and 526: "imidazothiadiazole sulfonamides" and "hydroxamamide" can be deleted because they are already described earlier.

We deleted "imidazothiadiazole sulfonamides" and "hydroxamamide" in lines 513 and 526.

22. Legends of Figure 11 and Figure 12 should end with ".".

It was changed accordingly.

23. Table 2. Model instead of "model".

We corrected it.

Hopefully, the reviewer will find that our answers to the comments are satisfying.